# The Persistence Length of Semiflexible Polymers in Lattice Monte Carlo Simulations

**DOI:** 10.3390/polym11020295

**Published:** 2019-02-10

**Authors:** Jing-Zi Zhang, Xiang-Yao Peng, Shan Liu, Bang-Ping Jiang, Shi-Chen Ji, Xing-Can Shen

**Affiliations:** State Key Laboratory for Chemistry and Molecular Engineering of Medical Resources, School of Chemistry and Pharmaceutical Sciences, Guangxi Normal University, Guilin 541004, China; jingzi_zhang@foxmail.com (J.-Z.Z.); pengxy_1026@163.com (X.-Y.P.); 2172226031@email.szn.edu.cn (S.L.); jiangbangping@mailbox.gxnu.edu.cn (B.-P.J.); xcshen@mailbox.gxnu.edu.cn (X.-C.S.)

**Keywords:** semiflexible polymer, persistence length, Monte Carlo simulation, bond fluctuation model, lattice simulation

## Abstract

While applying computer simulations to study semiflexible polymers, it is a primary task to determine the persistence length that characterizes the chain stiffness. One frequently asked question concerns the relationship between persistence length and the bending constant of applied bending potential. In this paper, theoretical persistence lengths of polymers with two different bending potentials were analyzed and examined by using lattice Monte Carlo simulations. We found that the persistence length was consistent with theoretical predictions only in bond fluctuation model with cosine squared angle potential. The reason for this is that the theoretical persistence length is calculated according to a continuous bond angle, which is discrete in lattice simulations. In lattice simulations, the theoretical persistence length is larger than that in continuous simulations.

## 1. Introduction

It is common to study polymers with stiffness, such as DNA, protein, filament, polyelectrolyte, molecular brush and liquid crystalline polymer. Usually, the stiffness of polymers is characterized by persistence length that can be understood as the distance over which the orientation of the bonds persists. Persistence length connects the studies of experiments, theories and simulations; it is a primary task to determine persistence length in studies of semiflexible polymers.

In the Kratky−Porod worm-like chain (WLC) model [1], which is the most successful theory of dealing with rigid and semiflexible polymers [2,3,4], the persistence length (*l_p_*) of a linear polymer in a three-dimensional space is defined by its relationship with the bending stiffness λ as $l_{p}=\lambda /k_{B}T$ (*k_B_T* is the thermal energy) [5,6]. The WLC model describes a polymer with a contour length *L* in terms of a smooth contour ***r***(*t*) in space, where *t* is a coordinate along the contour, and the bending energy is
(1)$$E=\frac{\lambda }{2}\int_{0}^{L}dt{\left(\frac{d^{2}r}{dt^{2}}\right)}^{2}$$
Here, the parameter λ describes the bending stiffness of the chain.

While applying molecular simulations, such as molecular dynamics (MD) simulation and Monte Carlo (MC) simulation, to study semiflexible polymers, the chain stiffness is usually introduced via a certain bending potential *E* between successive bonds and modulated by the bending constant *k* [7,8,9,10,11,12,13,14,15,16,17,18,19,20,21,22,23]. Then, two questions arise: How can persistence length be calculated, and what is the relationship between persistence length and the bending constant *k*; that is, what is the theoretical persistence length?

Persistence length can be calculated according to properties such as the orientation correlation of bond vectors, the average projection of end-to-end distance onto the first bond, the bond angle, etc. [4,9,10,11,24,25,26,27,28]. The calculation methods of persistence length have been well studied [9,10]. Hsu et al. emphasized that the frequently used exponential decay of the orientation correlation only holds for Gaussian phantom chains [10]. Furthermore, Cifra recommended that it is safer to estimate persistence length according to the projection of end-to-end distance onto the first bond [9]. Nowadays, it is accepted that the bond angle effectively characterizes the local stiffness of a linear polymer chain [9,10].

Theoretical persistence length is helpful in guiding simulations. WLC can be discretized as a polymer with monomers connected by cosine angle (CA) potential;
(2)$$E/k_{B}T=k\left(1-\cos\theta \right),$$
where *θ* is the bond angle between two successive bond vectors [6,29]. Theory predicts that the persistence length is $l_{p}\sim k$ [29,30,31], which has been confirmed in a MD simulation [10].

Due to their high computational efficacy, lattice MC simulations are widely used to study semiflexible polymers [5,6,7,8,9,10,11,12]. However, in simulations with a standard simple cubic lattice model (SCLM), in which the bond length is fixed as one unit of lattice spacing and the bond angle is 0 or π/2 (Figure 1a), the persistence length of a polymer with CA potential significantly deviates from the theoretical prediction and equals $l_{p}\sim \exp\left(k\right)/4$ [6,11]. Such a deviation is also found in other lattice simulations [9,10]. Hsu et al. suggested that it is due to the limited angles in lattice simulations [6,11], but the theoretical persistence lengths in other lattice models such as the Larson-type bond fluctuation model (L-BFM, Figure 1b) and the bond fluctuation model (BFM, Figure 1c) have not been analyzed.

Apart from the CA potential, cosine squared angle (CSA) potential [16,32,33,34,35]
(3)$$E/k_{B}T=k{\left(1-\cos\theta \right)}^{2},$$
is also a popular bending potential which is advantageous to use in dynamics simulations with a nonuniform curvature [36]. To our surprise, the theoretical persistence length with CSA potential has not been reported.

In this paper, we obtained theoretical persistence length for polymers with CA and CSA potentials and applied lattice MC simulations to study their persistence length. The outline of the paper is as follows: Section 2 introduces the deduction of theoretical persistence length, Section 3 describes the lattice Monte Carlo model and the calculation methods of persistence length and Section 4 studies the behavior of polymers with both bending potentials and analyzes the theoretical persistence length in lattice MC simulations. A brief conclusion is given in Section 5.

## 2. Theoretical Persistence Length

The mean square end-to-end distance of a semiflexible ideal polymer with chain length N≫1 can be expressed as [6,7,8,31,37]
(4)$$\langle R_{e}^{2}\rangle =Nl^{2}\frac{1+\langle \cos\theta \rangle }{1-\langle \cos\theta \rangle }=Nl_{k}l,$$
where *l_k_* is the Kuhn length. Since persistence length is equal to one-half the Kuhn length [6,8,29,30,31], it can be calculated as
(5)$$l_{p}=\frac{l}{2}\cdot \frac{1+\langle \cos\theta \rangle }{1-\langle \cos\theta \rangle }.$$

Thus the calculation of persistence length is converted to the calculation of mean cosine of bond angle 〈cosθ〉. When ignoring the excluded volume effect, 〈cosθ〉 of a semiflexible polymer with bending potential *E* can be calculated as [13,36,37,38]
(6)$$\langle \cos\theta \rangle =\frac{\int_{0}^{\pi }\cos\theta \sin\theta \exp\left(-E/k_{B}T\right)d\theta }{\int_{0}^{\pi }\sin\theta \exp\left(-E/k_{B}T\right)d\theta }.$$

For a semiflexible polymer with CA potential (Equation (2)), we can obtain [29,30]
(7)$$\langle \cos\theta \rangle =\frac{\left(k+1\right)\exp\left(-2k\right)-1+k}{k\left(1-\exp\left(-2k\right)\right)},$$
which is consistent with the literature [37]. Substituting into Equation (5), the popular expression of the persistence length of a worm-like chain $l_{p}=lk$ can be obtained when k≫1 [6,29,30].

When the monomers of a polymer are connected by CSA potential (Equation (3)), the mean cosine of bond angle is
(8)$$\langle \cos\theta \rangle =1-\frac{1-\exp\left(-4k\right)}{\sqrt{\pi k}\mathrm{erf}\left(2k\right)}.$$
The theoretical persistence length is $l_{p,th}=l{\left(\pi k\right)}^{1/2}$ when *k* >> 1. Clearly, the scaling relationship between persistence length *l_p_* and bending constant *k* is different for simulations with CA than it is for simulations with CSA potentials.

## 3. Models and Simulation Methods

### 3.1. Lattice MC Simulations

We applied the bond fluctuation model (BFM) [35,39] to study a semiflexible polymer with *N* bonds (N+1 monomer) in a simple cubic latttice with periodical boundary conditions. In BFM, each monomer occupies eight corners of a cube that are connected via one of 108 bond vectors of length 2, $\sqrt{5}$, $\sqrt{6}$, 3 or $\sqrt{10}$ in units of lattice spacing, in 87 possible bond angles (Figure 1c). Thus, BFM is a good approximation of a continuum simulation, and maintains the advantages of lattice simulations. The microrelaxation mode adopts the traditional “L6” move: A monomer can randomly move one unit in one of the six directions. The move is accepted only when the excluded volume effect and bond length constraint are not violated. Wittmer et al. have suggested a L26 move [40]: One monomer can move to one of its 26 destinations, which dramatically speeds up the dynamics at the expense of bond cross.

For comparison purposes, we also applied a Larson-type bond fluctuation model (L-BFM) [41,42,43], in which each monomer occupied one lattice with a permitted bond length of 1 or $\sqrt{2}$ (Figure 1b). During relaxation, each monomer could move to one of 18 destinations (nearest and next-nearest neighbors).

Two types of bending potentials were examined: One was cosine angle (CA) potential (Equation (2)), which could be deduced from the discretized WLC model [6,29]. The other was cosine squared angle (CSA) potential (Equation (3)), which was proposed by Binder in simulations with BFM [35].

In this simulation, the excluded volume effect was considered; each lattice site could only be occupied once and no bond cross was allowed. The time unit was MC step (MCS), during which all monomers attempted to move once on average. A metropolis importance algorithm was employed [44]; one attempted movement was accepted with a probability $P=min\left\{1,\exp\left(-\Delta E/k_{B}T\right)\right\}$, where Δ*E* is the energy change due to the movement and *k_B_T* was set to be 1.

### 3.2. Calculation of Persistence Length

There are numerous methods of calculating persistence length (*l_p_*) in simulations and experiments [4,9,10,11,24,25,26,27,28]. Persistence length (*l_p_*) is usually obtained from the exponential decay of the orientation correlation of bond vectors along the chain
(9)$$\langle r_{i}\cdot r_{i+s}\rangle =l^{2}\langle \cos\theta \left(s\right)\rangle =l^{2}\exp\left(-sl/l_{p}\right).$$
Here, *s* is the chemical distance between two bonds, and *l* is the bond length. In BFM and L-BFM, the average bond length 〈*l*〉 is used, which is slightly affected by the chain stiffness [11].

When Equation (9) holds, the persistence length can be directly calculated from the bond angle (*s* = 1) [10,11,12]
(10)$$l_{p}=-\langle l\rangle /\ln\left(\langle \cos\theta \rangle \right).$$
Hsu et al. emphasized that this method is better than that based on the orientation correlation [10]. 

According to Flory, persistence length is the average projection of end-to-end distance onto the first bond of the chain [9,45,46]. Thus, it can be calculated as the sum of projection of all bond vectors onto the first bond vector
(11)$$l_{p}=\langle \overset{N}{\underset{i=1}{\sum }}\vec{l_{i}}\;\vec{l_{1}}\rangle /\langle l\rangle .$$
This method does not depend on the chain model and state of chains [9]. However, the result is affected by the chain length and cannot exceed the contour length of the chain.

According to the WLC model, the mean square end-to-end distance of a WLC with contour length L=Nl is
(12)$$\langle R_{e}^{2}\rangle =2l_{p}L\left\{1-\frac{l_{p}}{L}\left[1-\exp\left(-L/l_{p}\right)\right]\right\}.$$
Thus, the persistence length can be deduced from $\langle R_{e}^{2}\rangle$ and *L*. This method has been adopted to estimate the persistence length of DNA in experiments [25,28].

The aforementioned methods of calculating persistence lengths are based on the orientation correlation of bond vectors (Equation (9)), the cosine bond angle (Equation (10)), the projection of end-to-end distance (Equation (11)) and the mean square end-to-end distance of WLC (Equation (12)). For convenience, the corresponding persistence lengths are marked as *l_p,auto_*, *l_p,θ_*, *l_p,ee_* and *l_p,WLC_*, respectively.

## 4. Results and Discussion

### 4.1. BFM with CA Potential

Firstly, we applied BFM to examine the properties of a polymer chain with N=50 monomers connected by CA potential. Such a chain length is long enough to ensure that the obtained persistent length is not affected by the chain length in the examined range of bending constant [9]. The effects of chain stiffness on polymer size are shown in Figure 2. Both the mean square end-to-end distance $\langle R_{e}^{2}\rangle$ and the mean square radius of gyration $\langle R_{g}^{2}\rangle$ increased by increasing the bending constant *k*. When k=0, the ratio $\langle R_{e}^{2}\rangle /\langle R_{g}^{2}\rangle =6.4$ was consistent with the result of a flexible self-avoiding chain [10]. The ratio increased to 10.5 when k=20, which suggests that the chain gradually changed from a random coil to a rodlike structure. The mean cosine of the bond angle 〈cosθ〉 gradually increased by increasing the bending potential *k*, suggesting that successive bonds oriented in the same direction (Figure 2b).

The orientation correlation 〈cosθ(s)〉 as a function of the chemical distance *s* between two bonds was examined (Figure 3). As expected, ln(〈cosθ(s)〉) decreased linearly by increasing *s*, indicating an exponential decay of the bond orientation correlation. The orientation correlation decayed faster by decreasing *k* and became ill-defined for small k (k<5), which has been observed in rather flexible self-avoiding chains in the literature [11].

We compared the persistence lengths obtained from different calculation methods. As shown in Figure 4, all the persistence lengths increased by increasing the bending constant *k*. No significant difference could be observed for small k (k<7), and all the curves were almost identical, which is consistent with the literature [9]. The persistence length *l_p,θ_*, which was obtained from the cosine bond angle, increased more slowly than other lengths with further increasing *k*. When k>13, the persistence length *l_p,auto_*, obtained from the bond−bond orientation correlation showed a rapid increase, and was then followed by *l_p,WLC_* (according to the mean square end-to-end distance of WLC) and *l_p,ee_* (according to the projection of end-to-end distance). In the end, the relative relationship was *l_p,auto_ > l_p,WLC_ > l_p,ee_ > l_p,θ_* and all simulated persistence lengths were larger than theoretical prediction *l_p,th_.* Both *l_p,auto_* and *l_p,WLC_* exceeded the contour length of the chain *L* at high chain stiffness (k>19).

In simulations, it is important to ensure that persistent length is not affected by chain length. We further examined polymers with N=20 and 70, respectively (Figure 5). At low chain stiffness (*k* < 10), persistent length was not affected by chain length, which is consistent with the conclusion made by Cifra [9]. At high chain stiffness, the result was affected by both the chain length and the calculation methods. The calculation of *l_p,ee_* was suitable for all chain models [9], but *l_p,ee_* was even smaller than *l_p,th_*, while the result of N=20 and the result of N=70 were larger than that of N=50. Thus, we do not recommend calculating *l_p,ee_* in simulations as it is significantly affected by chain length. For both *l_p,auto_* and *l_p,WLC_*, the results of N=50 and 70 were almost identical and were larger than that of N=20, which suggests that N=50 was long enough in the examined range of chain stiffness [9]. It is interesting that *l_p,θ_* was not affected by chain length; thus, we recommend calculating *l_p,θ_* in simulations [9,10]. Among all the calculation methods, *l_p,θ_* reflects local stiffness whereas other methods reflect the overall stiffness of a chain. As expected, *l_p,θ_* corresponds to *l_p,th_* well as both are calculated according to 〈cosθ(s)〉. However, attention should be paid so that *l_p,θ_* deviates from *l_p,th_* at high chain stiffness; for example, that the relative deviation $\left(l_{p,\theta }-l_{p,th}\right)/l_{p,th}$ is 26.2% when $l_{p,th}/l=18$. Such a deviation has also been observed for a semiflexible polymer with $N=10^{4}$ studied by BFM (the results are shown as the open squares in Figure 5) [11].

### 4.2. BFM with CSA Potential

When CSA potential was applied, the corresponding bending constant *k* was in the range of 0~120 in order to obtain similar persistence length as the simulation with CA potential. The mean square end-to-end distance $\langle R_{e}^{2}\rangle$, mean square radius of gyration $\langle R_{g}^{2}\rangle$ and their ratio increased by increasing the bending constant *k* (Figure 6a). The mean cosine angle 〈cosθ(s)〉 increased by increasing *k*, which was consistent with the theoretical prediction (Figure 6b). Thus, the chain stiffness was successfully introduced.

Figure 7 shows persistence lengths obtained by different methods in the simulation with CSA potential; these share some similarities with the simulation results with CA potential (Figure 4). Different methods obtained similar persistence lengths at low chain stiffness (k<20). When the stiffness was high, a divergence was observed with the relationship as *l_p,auto_ > l_p,WLC_ > l_p,ee_ > l_p,θ_*. Within the examined range of bending constant *k*, *l_p,θ_* was consistent with the theoretical prediction $l{\left(\pi k\right)}^{1/2}$ and the relative deviation $\left(l_{p,\theta }-l_{p,th}\right)/l_{p,th}$ was 4.0% when $l_{p,th}/l=18$, which was much smaller than that with CA potential.

### 4.3. Theoretical Persistence Length in Lattice Simulations

From the above results, it can be concluded that BFM with CSA potential obtains a persistence length *l_p,θ_* that is consistent with the theoretical prediction *l_p,th_* and is much better than that with CA potential. As far as persistence length is concerned, MC simulations with BFM should adopt CSA potential to study semiflexible polymers.

Carefully examining the deduction of theoretical persistence length, we can find that it is calculated from a continuous bond angle (Equation (6)), which is discrete in lattice simulations [6,11]. The mean cosine of a bond angle in its discrete form is
(13)$$\langle \cos\theta \rangle =\frac{\sum f_{i}\cos\theta \exp\left(-E_{i}/k_{B}T\right)}{\sum f_{i}\exp\left(-E_{i}/k_{B}T\right)}.$$
Here, *f_i_* is the priori probability of the bond angle *θ_i_* for a chain consisting of two bonds only [39], which is determined by the simulation model. There are 87 different bond angles in BFM with the L6 move [39]. The probability of a bond angle has a maximum θ=π/2 and is roughly symmetric around this value (Figure 8) [39].

Figure 9a compares the theoretical persistence length in BFM (*l_p,BFM_*) and continuous simulations (*l_p,th_*) with CA potential. At low chain stiffness (k<15), *l_p,BFM_* corresponds to *l_p,th_* very well. However, at high chain stiffness (40<k<100), *l_p,BFM_* is much larger than *l_p,th_* and obeys a scaling law $l_{p,BFM}\sim k^{2.3}$. When CSA potential is applied, the theoretical persistence length *l_p,BFM_* is $l_{p,BFM}\sim {\left(\pi k\right)}^{1/2}$ at low chain stiffness (k<100) and increases with the bending constant *k* at a higher chain stiffness with a linear relationship $l_{p,BFM}/l\approx 7.2+0.1k$ (Figure 9b). In the examined range of *k*, the difference between *l_p,BFM_* and *l_p,th_* is small while CSA potential is applied. For both CA and CSA potentials, the simulation results *l_p,θ_* agree with *l_p,BFM_* better than *l_p,th_*.

To examine the influence of simulation models, we studied polymers with a Larson-type bond fluctuation model (L-BFM), in which the bond length was 1 or $\sqrt{2}$ with six different bond angles (Figure 1b). The theoretical persistence lengths in L-BFM are $l_{p,LBFM}∼\exp\left(0.29k\right)$ for CA potential and $l_{p,LBFM}∼\exp\left(0.083k\right)$ for CSA potential (Figure 10). The simulation results are identical with the theoretical predictions for both CA and CSA potentials (Figure 10).

When CA potential was applied, theoretical persistence length *l_p_* at high chain stiffness was: $l_{p}∼k$ with continuous bond angles, $l_{p}∼k^{2.3}$ in BFM with 87 bond angles, $l_{p}∼\exp\left(0.29k\right)$ in L-BFM with six bond angles and $l_{p}∼\exp\left(k\right)$ in SCLM with two bond angles. Clearly, *l_p_* increased rapidly when the number of permitted bond angles decreased.

## 5. Conclusions

Theoretical persistence length can be applied to estimate persistence length and judge the correction of simulation results. In this paper, the theoretical persistence lengths of linear polymers with CA and CSA potentials were deduced from bond angles which equaled lk and $l{\left(\pi k\right)}^{1/2}$, respectively.

We applied BFM to study semiflexible linear polymers and calculated persistence length with different methods. It was found that *l_p,θ_*, persistence length calculated according to the cosine of bond angle was close to the theoretical prediction *l_p,th_*. Nevertheless, a significant deviation between *l_p,θ_* and *l_p,th_* could be observed at not high chain stiffness (k>7) when CA potential was applied. For CSA potential, the deviation was small when k<120 and the expression $l{\left(\pi k\right)}^{1/2}$ could be used to roughly estimate persistence length.

The simulation result *l_p,θ_* in BFM did not match the theoretical prediction *l_p,th_*, mainly because *l_p,th_* is deduced from continuous bond angles which are discrete in BFM. The theoretical persistence length in lattice simulations was affected by both the type of lattice models (BFM, L-BFM or SCLM) and bending potentials (CA or CSA potential). Only in BFM with CSA potential, the theoretical persistence length with continuous bond angles could roughly estimate persistence length. For other cases, the simulated persistence length should be compared with the theoretical persistence length in corresponding lattice simulations. This study is helpful in understanding the difference between the Kratky-Porod off-lattice model and various lattice models for semiflexible polymers.

## Figures and Tables

**Figure 1 polymers-11-00295-f001:**
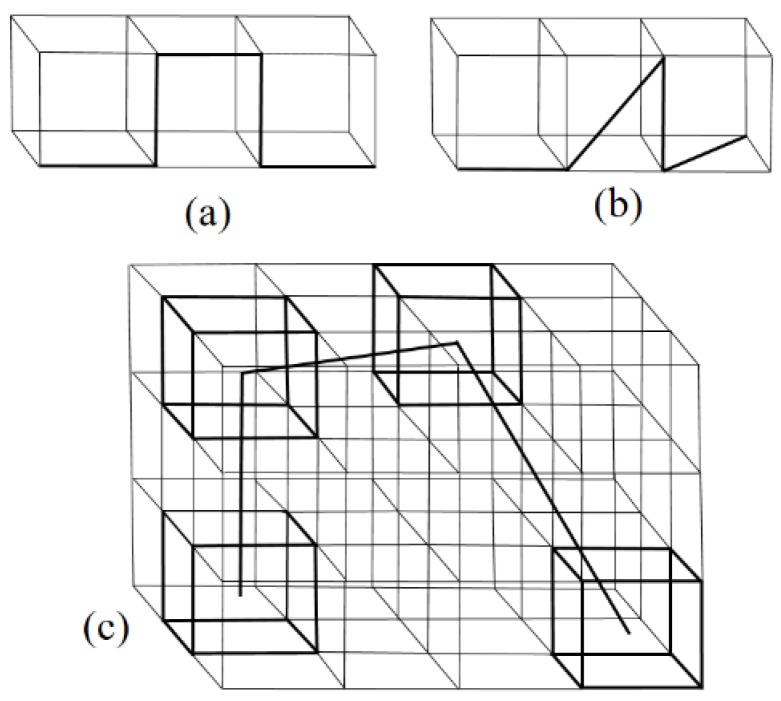
The illustration of lattice Monte Carlo simulations: (**a**) A simple cubic lattice model (SCLM); (**b**) a Larson-type bond fluctuation model (L-BFM); (**c**) a bond fluctuation model (BFM).

**Figure 2 polymers-11-00295-f002:**
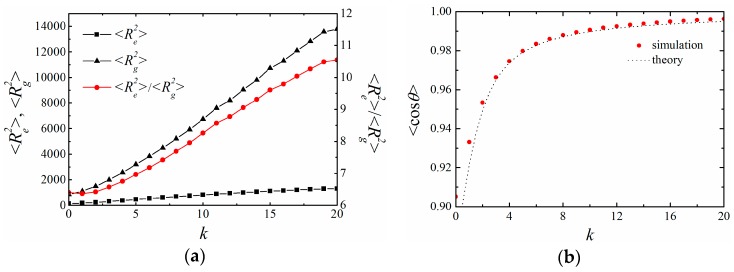
Effect of the bending constant *k* in BFM with cosine angle (CA) potentials. (**a**) The mean square end-to-end distance $\langle R_{e}^{2}\rangle$, radius of gyration $\langle R_{g}^{2}\rangle$ and their ratio; (**b**) the mean cosine bond angle 〈cosθ〉, where the dash line is the theoretical value. The length of the chain is N=50. All results are averaged over 100 independent runs.

**Figure 3 polymers-11-00295-f003:**
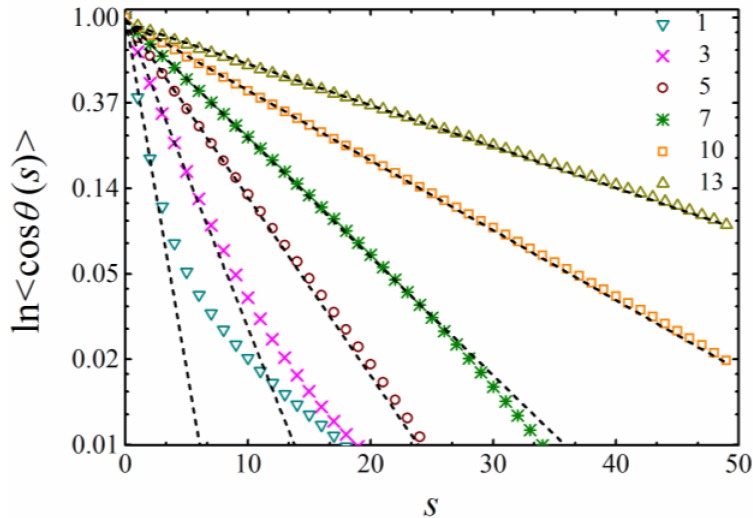
The orientation correlation function as a function of the chemical distance *s* along the chain with an indicated bending constant *k*. The dashed lines indicate the fits of the initial decay.

**Figure 4 polymers-11-00295-f004:**
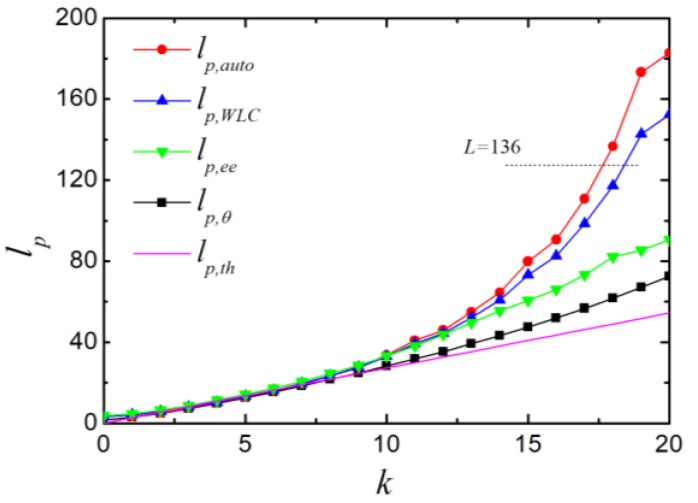
The persistence lengths as a function of bending constant *k*. *l_p,auto_*, *l_p,WLC_*, *l_p,ee_* and *l_p,θ_* are persistence lengths calculated according to the orientation correlation function, the mean square end-to-end distance of a worm-like chain, the projection of the end-to-end vector on the first bond and the bond angle, respectively. The contour length *L* of the chain is 136.

**Figure 5 polymers-11-00295-f005:**
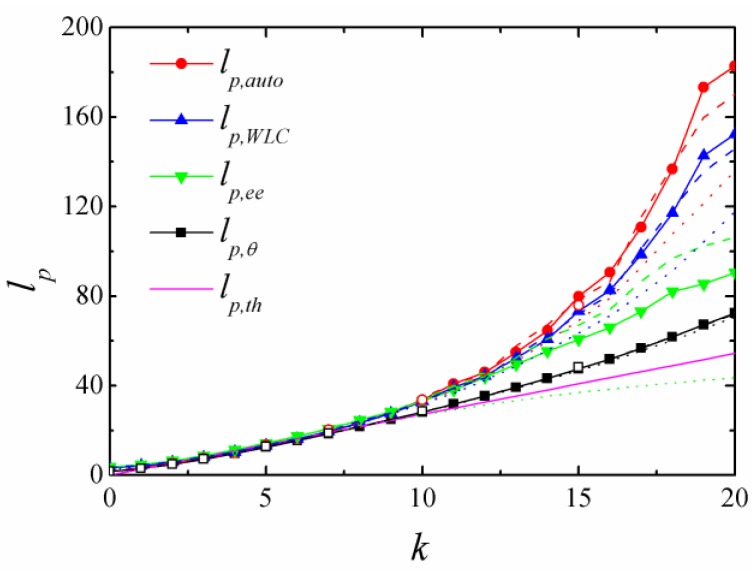
Various persistent lengths obtained with different chain lengths: *N* = 20 (dot line), 50 (solid line and symbol) and 70 (dash line). The open squares and spheres are the data from table IV in Ref. [11] which were calculated according to bond angle and orientation correlation for a polymer with $N=10^{4}$.

**Figure 6 polymers-11-00295-f006:**
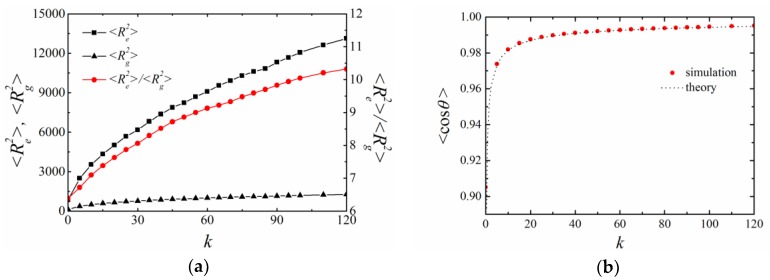
The effect of bending constant *k* on (**a**) $\langle R_{e}^{2}\rangle$, $\langle R_{g}^{2}\rangle$ and their ratio, (**b**) the mean cosine of bond angle 〈cosθ〉. CSA potential is applied in BFM, the others parameters are the same as those in Figure 2.

**Figure 7 polymers-11-00295-f007:**
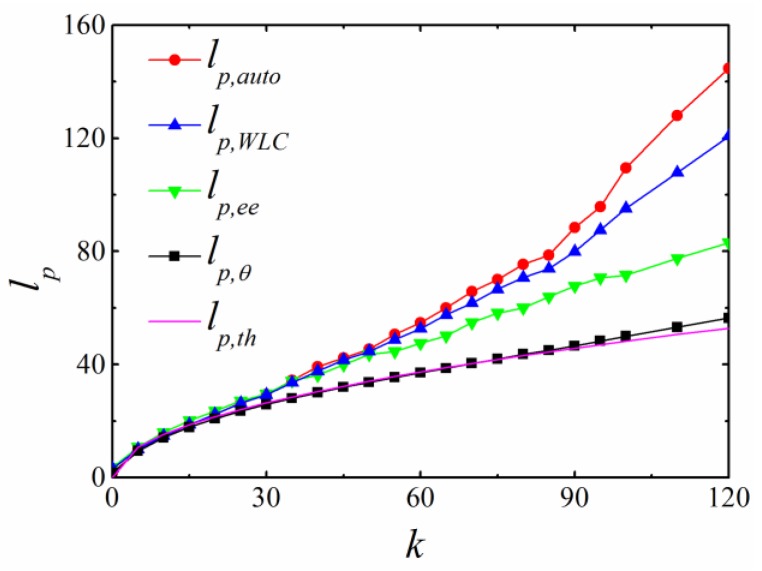
The persistence lengths as a function of bending constant *k* when CSA potential was applied in BFM.

**Figure 8 polymers-11-00295-f008:**
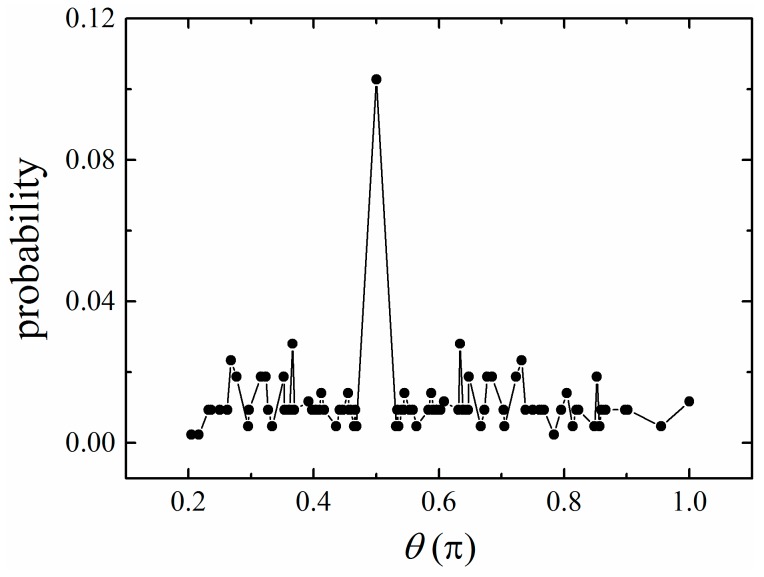
The priori probability of the angle between two successive bond vectors in the bond fluctuation model (BFM). The angle is measured in units of π.

**Figure 9 polymers-11-00295-f009:**
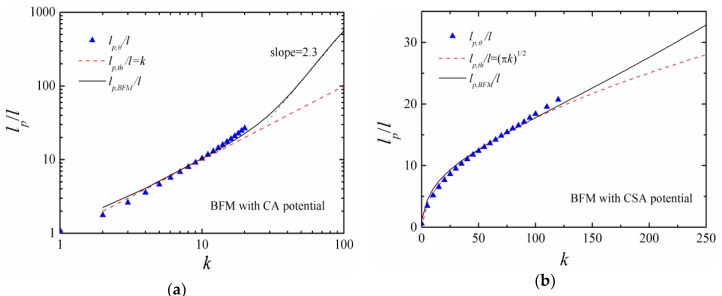
The theoretical persistence lengths in BFM and continuous simulations and simulation result *l_p,θ_* with (**a**) CA and (**b**) CSA potentials, respectively.

**Figure 10 polymers-11-00295-f010:**
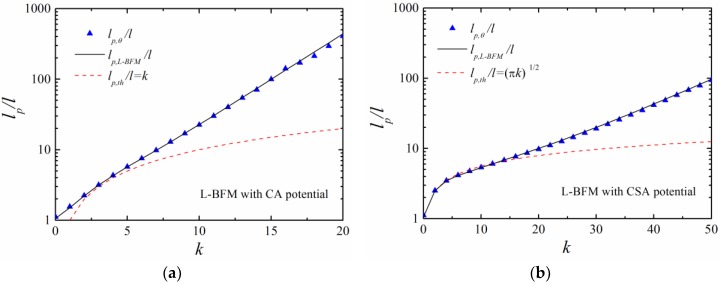
The theoretical persistence lengths in Larson-type BFM and continuous simulations and simulation results *l_p,θ_* with (**a**) CA and (**b**) CSA potentials, respectively. The length of the chain is *N* = 50.

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
