# Peer review of "The Persistence Length of Semiflexible Polymers in Lattice Monte Carlo Simulations"

_polymers, 2019, doi:10.3390/polym11020295_

Round 1
Author Response
Thank you very much for such a positive comment to the novelty and scientific value of our paper. Your suggestion does help us to improve our manuscript. More details please check the attack file.
Best regards,
Shichen JI

Reviewer 2 Report
This manuscript by Zhang et al. describes theoretical approach to estimating persistence length of polymers. The authors examined two types of bending potentials combined with Monte Carlo simulation. The theoretical foundation is stiff and sound, and the results are well discussed in detail. I would recommend this paper for publication as is.
Author Response
Thank you very much for your kind consideration.
Best regards,
Shichen Ji